# Team-Assisted Individualization Type of the Cooperative Learning Model for Improving Mathematical Problem Solving, Communication, and Self-Proficiency: Evidence from Operations Research Teaching

**Georgina Maria Tinungki [1,*], Budi Nurwahyu [2], Agus Budi Hartono [3] and Powell Gian Hartono [4]**

1    Department of Statistics, Faculty of Mathematics and Natural Sciences, Hasanuddin University, Makassar 90245, Indonesia
2    Department of Mathematics, Faculty of Mathematics and Natural Sciences, Hasanuddin University, Makassar 90245, Indonesia
3    Department of Ship Machinery, AMI Maritime Polytechnic, Makassar 90134, Indonesia
4    Master of Management Program, Satya Wacana Christian University, Salatiga 50711, Indonesia
*    Correspondence: georgina@unhas.ac.id

**Abstract:** The purpose of this study is to analyze the increase in the mathematical problem-solving (MPS), mathematical communication (MC), and self-proficiency (SPr) abilities of students by applying the team-assisted individualization type (TAI) of the cooperative and conventional learning models for the experimental and control classes. This is a quasi-experimental study comprising a sample of 50 and 42 students studying for an undergraduate degree in statistics for the experimental and control classes, respectively. Students' mathematical prior knowledge (MPK) is grouped into three levels, namely high, medium, and low. The instruments used are the MPS test, the MC test, the self-proficiency scale, and observation sheets. Statistical analysis instruments used are parametric and non-parametric statistics, such as prerequisite tests for normality and homogeneity of variance, mean difference tests in two or more groups, the post hoc test, and description and interaction tests. The results showed an increase in MPS, MC, and SPr in the experimental class, which is higher than in the control class. Furthermore, there is no interaction between the experimental and control classes despite a significant correlation between MPS, MC, and MPS. These findings are relevant to mathematics and statistics teaching because it has been proven to improve students' MPS, MC, and SPr; hence, learning outputs can be achieved objectively, specifically for operations research teaching.

**Keywords:** team-assisted individualization (TAI); cooperative learning model; mathematical problem solving; mathematical communication; self-proficiency; operations research

## 1. Introduction

Mathematics learning is not only aimed at ensuring students have optimal understanding of the material; it also aids to improve their communication abilities, engagement, representation, and problem-solving skills. Moreover, students can imbibe the skills and abilities they use for mathematics and internalize them so they can use them outside the classroom Mathematics learning is not only aimed at ensuring students have optimal understanding of the material; it also aids to improve their communication abilities, engagement, representation, and problem-solving skills. Moreover, students can imbibe the skills and abilities they use for mathematics and internalize them so they can use them outside the classrooms [1]. According to the National Council of Teacher Mathematics (NCTM), the standard for learning this subject depends on the student's ability to communicate, solve problems, make connections, and argue [2]. Students with high learning motivation can possess great mathematical reasoning ability [3,4].

Students' reticence towards mathematics can pose a challenge. Mathematics is the "queen of science" for developing science and technology, and has many everyday applications [5]. The importance of learning mathematics applies to all aspects of life; all branches of science require mathematics that suits their needs. Mathematics teaches students effective, concise, and clear means of communication and gives them the skills to present information in a variety of ways. It also can improve their ability to think logically and accurately, as well as raise their spatial awareness, and it gives them the satisfaction of solving challenging problems [6]. Furthermore, mathematical problem-solving and communication are interpreted as teaching methods that can improve the quality of mathematics teaching in educational institutions, as well as for higher education [1,7].

Based on initial observations of the Operations Research course at the undergraduate program of Statistics, it was reported that the majority had poor mathematical problem-solving, communication, and self-proficiency abilities. The low understanding of the subject matter can be attributed to the learning process' focus on the passivity of students who only rely on formulated materials. It is also in line with the strategies provided by the lecturers without studying independently to have a more comprehensive understanding. Lecturers also did not allow students to construct mathematical knowledge that students will have more. In addition, students tended not to be proficient with practice questions. Their ability to understand mathematical problems logically was low, they had a low ability to understand concepts from lecture material, and they were weak in understanding procedures in working on mathematical problems given in class. An indication of low problem-solving ability is their failure to work on enrichment, which is entirely different from the assigned questions. Meanwhile, an indication of low mathematical communication ability is their incapability to explain the problem-solving steps. Moreover, the indication of low self-proficiency in students is low proficiency in understanding mathematical problems and concepts, and low understanding of procedures for solving mathematical problems. This becomes the benchmark for a more comprehensive analysis [8–12].

Mathematics problem-solving involves understanding the question, implementing completion plans, and evaluating work results [13]. Irrespective of the fact that these stages are incomplete, they are interconnected. Students with a low understanding of the material tend to have difficulties engaging in mathematical problem-solving processes [8]. Preliminary studies on improving its aspects were carried out by [7,10,14].

Students must develop effective mathematical communication skills to curb the difficulties usually encountered in understanding the learning materials [15]. The components of mathematical communication include use of appropriate terms, understandable conversation, transition to cues, emphasis factors, and behavioral adjustments between verbal and non-verbal languages [16]. Its indicators involve having a clear discussion with lecturers and other students, applying mathematical ideas and languages appropriately, as well as analyzing and evaluating proposed strategies [17].

Mathematical ideas are usually communicated during the learning process in the classroom. Interestingly, mathematical language improves the student's communication abilities [18]. Preliminary studies further reported that it is also enhanced by group discussion [19]. Research on the weaknesses of mathematical communication was carried out by [1,20–22]. Previous studies stated that learning is needed to improve this skill alongside problem-solving abilities.

One alternative to improving mathematical problem-solving and communication abilities is the application of cooperative learning. This process trains students to listen to other people's opinions and draw conclusions [23]. It also provides opportunities for students from diverse backgrounds to work interdependently in terms of completing joint tasks and to learn mutual respect. In addition to improving social skills, cooperation and collaboration abilities are obtained from this learning model [1].

However, another aspect related to the students' affective domain is self-proficiency, a component of learning independence [24]. It is defined as the capability to understand as well as to apply the appropriate procedures and strategies when solving a mathematical

problem. This ability determines the outcomes even before the action occurs [25]. An empirical study on self-proficiency was carried out by [12], although it only examines the achievement of learning outcomes rather than their increase.

Self-proficiency is different from self-efficacy, as reported by [25]. Based on Bandura's socio-cognitive theory, self-efficacy depends on the proposition that a person's achievement or performance depends on the behavioral interaction between personal factors and environmental conditions [26]. It is also described as the belief in one's ability to organize and implement the specific actions necessary to manage prospective situations. Self-efficacy expectations are a person's assessment of the relevant skills, actions, or perseverance required for a given outcome [27].

Furthermore, self-proficiency increases self-confidence, which is key in the ability to solve mathematical problems. Students are expected to possess mathematical problem-solving and communication skills, as well as self-proficiency skills, to solve questions, interact reasonably, illustrate specific ideas into models, and connect what they are learning with other concepts and disciplines [12].

Some difficulties associated with mathematical problem-solving, communication, and self-proficiency abilities were bridged by the team-assisted individualization (TAI) type of cooperative learning model. This is a learning process executed in small heterogeneous groups. The interactive session helps participants understand the learning material and boosts positive interdependence rules, individual responsibility, intensive face-to-face communication, and group process evaluation. This process has certain implications for effective classroom management. The application of the TAI type of the cooperative learning model is adjusted based on the student's characteristics and needs during the learning process [1,28].

Based on the afore-mentioned description, the present study was carried out to examine whether there is an improvement in students' mathematical problem-solving, communication, and self-proficiency abilities through the TAI type of the cooperative learning model. The study was conducted using quasi-experimental methods with a quantitative approach to 92 undergraduate students in statistics who took operations research as a mathematics course; thus, research in the context of mathematics teaching was fulfilled. It addresses a gap in the literature by identifying the aspect of self-proficiency, which is rarely studied. The study of improving the aspect of self-proficiency with other aspects, such as mathematical problem solving and communication, was proven in applying the team-assisted individualization type of the cooperative learning model. In addition, it was necessary to prove the correlation between the three variables, and this analytical technique measures the strength of the relationship between two or more variables [29].

The object of this study is operations research, a compulsory general skills course in the undergraduate program of statistics. The main course materials include linear programming, simplex, transportation methods, queuing analysis, game theory, and CPM-PERT. Students need to master this course to understand other mathematical fields, specifically in the engineering and operations management departments. The operations research course requires mathematical problem solving, communication, and self-proficiency abilities. Knowledge of calculus and basic statistics is also needed to absorb all the presented materials.

This study makes several contributions: First, it examines the increase in mathematical problem-solving ability through the TAI type of the cooperative learning model. Second, it examines the increase in mathematical communication ability through the TAI learning model. Third, this study examines the increase in self-proficiency ability through the TAI learning model. Fourth, it examines the correlation between mathematical problem solving, communication, and self-proficiency through the TAI learning model. The results show an increase in mathematical problem solving, communication, and self-proficiency in the experimental class where the TAI learning model was applied compared to the control. Furthermore, there is a correlation between these variables, meaning that the TAI learning model increases them.

## 2. Literature Review

### 2.1. Mathematical Problem Solving

Mathematical problem-solving has an abstract meaning based on various complementary theoretical approaches [30]. The cognitive function needed in learning mathematics is problem-solving. Problem-solving is a relevant aspect of the international assessment framework, as in TIMSS, PISA, and NAEP. This increases knowledge and skills to deal with problems in everyday life [31,32]. In higher education, mathematical problem-solving ability is needed to achieve optimal learning outcomes. In operations research courses, knowledge of calculus and basic statistics is needed to absorb all the presented materials. The importance of problem-solving abilities, as stated by [33], are as follows: (1) it is the primary goal of teaching mathematics, (2) the methods, procedures, and strategies applied are the core and main processes in the curriculum, and (3) it is also a fundamental skill in learning this subject. As an implication of the opinion mentioned above, problem-solving ability needs to be possessed by all mathematics students from the elementary to higher education levels.

The mathematical problem-solving abilities that need to be developed are (1) understanding mathematical concepts and terms, (2) emphasizing similarities, differences, and analogies, (3) identifying essential elements and selecting the appropriate procedures, (4) distinguishing between unrelated items, (5) carrying out assessment and analysis, (6) visualizing and interpreting quantity or space, (7) generalizing based on several examples, (8) changing known methods, and (9) having sufficient self-confidence and enthusiasm for the material [34]. In line with this perspective, some constructivists stated the importance of preparing students to be able to solve problems under uncertain or ambiguous situations [35]. An essential mathematical activity is the exploration of the problem openly, as well as expanding and developing the problem to investigate more results and more general problems [36,37].

The students' mathematical problem-solving ability was assessed using solved questions. In this case, the work on the questions and the evaluation process was carried out comprehensively. Therefore, this variable is interpreted as a basic ability that needs to be mastered by students, and they should also be allowed to develop their knowledge actively [3,4].

The guiding steps proposed by [13] are known as heuristic strategies. These are widely used as a reference by many people in terms of solving mathematical problems. Study the diagram in Figure 1 [13].

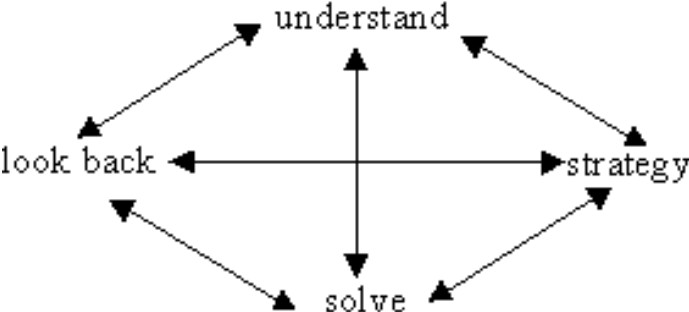

**Figure 1.** Mathematical Problem-Solving Stages.

Figure 1 shows the four-stage problem-solving model proposed by [13], involving (1) understanding and exploring the problem; (2) searching for strategies; (3) using these strategies to solve problems; and (4) looking back and reflecting on the solution [38]. The process of accessing each item of the mathematical problem-solving ability test refers to the holistic assessment or scoring, as shown in Table 1.

**Table 1.** Mathematical Problem-Solving Ability Scoring Rubric.

| No. | Identifying Known Elements | Implementing Strategies to Solve Problems | Explaining and Interpreting the Results |
|---|---|---|---|
| 1 | No element identification | No problem-solving strategy | Explanation and interpretation exist but are wrong |
| 2 | Element identification is incomplete | Problem-solving strategy is incomplete | Explanation and interpretation exist but are wrong and incomplete |
| 3 | Element identification is correct but incomplete | Problem-solving strategy is correct but incomplete | Explanation and interpretation are incomplete |
| 4 | Element identification is complete and correct | Problem-solving strategy is complete and correct | Explanation and interpretation are complete and correct |
| 5 | Maximum Score 4 | Maximum Score 4 | Maximum Score 4 |

Source: Modification from [39,40].

### 2.2. Mathematical Communication

Communication is a way of sharing ideas and clarifying one's understanding. However, these are perceived as objects of reflection, refinement, discussion, and reform. The communication process also helps to develop meaningful permanence for ideas and make them known to the public [2]. An educator must communicate effectively with students, colleagues, the administration, parents, and the outside community [41].

The students' mathematical communication ability needs to be developed because it helps students to express their thoughts orally and in written form. They can also respond appropriately to their fellow students and lecturers during the learning process [42]. Meanwhile, [2] stated that the mathematics learning program should provide the following opportunities for students to: (1) organize and consolidate their thoughts and ideas through communication, (2) communicate their thoughts logically and clearly to friends, teachers, etc., (3) analyze and evaluate other people's thoughts, and (4) use mathematical language to express their ideas appropriately.

There are two reasons why mathematical communication needs to be inculcated in students [1,43]. The first is that mathematics, as a language, does not only aid in thinking, finding patterns, solving problems, and drawing conclusions, but it is a valuable tool for communicating various ideas precisely and carefully. This subject symbolizes a series of meaningful statements conveyed by people. Mathematical symbols are artificial and only become significant after they have been defined. The second is mathematics learning is a social activity, meaning it acts as a vehicle that generates interaction among students and lecturers. Therefore, mathematical communication as a social activity (talking) and a thinking tool (writing) is recommended by experts to continue developing and improving among learners [44].

The indicators of mathematical communication, according to [45], are (1) the ability to express numerical ideas in both oral and written forms, as well as demonstrate and visually describe them, (2) understanding, interpreting, and evaluating these ideas orally or in other visual forms, and (3) the ability to use terms, arithmetic notations, and their structures to present ideas as well as to describe relationships and situation models. Furthermore, Ref. [46] stated that communication ability is essential, especially when discussions are being held among students. They are expected to be able to state, explain, describe, listen, ask, and cooperate to have a deeper understanding of the subject.

Aspects of this variable include how to develop students' mathematical communication ability to explain certain concepts or ideas. This is realized by providing opportunities for them to express themselves using pictures and signs as well as training them to relate mathematical problems to daily activities [1].

### 2.3. Self-Proficiency

In addition to the mathematical problem-solving and communication abilities included in the cognitive domain, students should also acquire skills related to the affective domain, such as self-proficiency [47]. Some experts defined this variable as a person's

capabilities, which are the competencies and skills possessed by individuals concerning their understanding and the adoption of procedures and strategies in certain circumstances. Students need to be adequately equipped with self-proficiency, boosting their confidence to face and solve life's challenges in general, or mathematical tasks in particular.

Self-proficiency is an ability, or competence, possessed by "self" relating to one's understanding, as well as the adoption of specific procedures and strategies in some circumstances. According to [48], its components consist of (1) conceptual understanding, (2) procedural fluency, (3) strategic competence, (4) adaptive reasoning, and (5) productive disposition. These five self-proficiencies should be coherent, and the strands need not be separated as they are intertwined into one functional ability in practice.

The five components of self-proficiency are described by [49] as follows:

Indicators of self-proficiency, such as procedural fluency, conceptual understanding, strategic competence, adaptive reasoning, and productive disposition, are described in Figure 2. Conceptual understanding is the students' comprehension or mastery of mathematical concepts, operations, and relations. Sub-indicators that can be used to determine whether they already possess this attribute are their ability to (a) restate the concepts studied, (b) classify objects based on the fulfillment or non-fulfillment of the requirements to form the model, (c) provide examples or non-examples of the topics studied, (d) present ideas in various forms of mathematical representation, (e) link various notions, and (f) develop the necessary and or sufficient conditions for the conceptions. According to [48], a significant indicator of this variable is the ability to present mathematical situations in different ways and to know how diverse representations can be used for various purposes. To find the solution to a problem, individuals need to ascertain how these representations are related and how they differ from one another. The students' conceptual understanding level relates to the connections they can make.

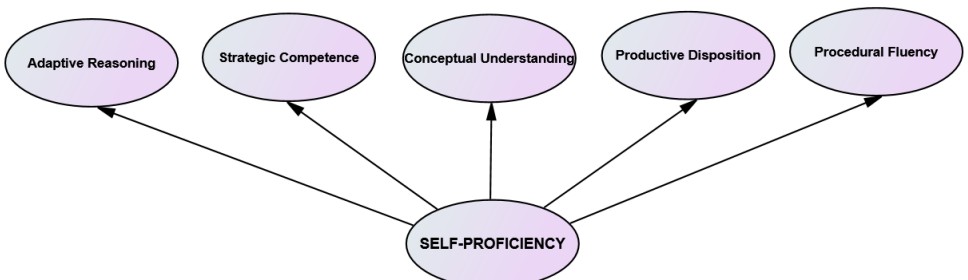

**Figure 2.** Self-Proficiency Component Correlation.

Procedural fluency refers to knowledge of procedures: when and how to use them appropriately, as well as how to utilize skills flexibly, accurately, and efficiently. Sub-indicators are the students' ability to (a) apply, (b) utilize, (c) select, (d) predict the outcome, (e) modify or refine, and (f) develop procedures. By studying algorithms as a "general procedure," students learn that mathematics is structured, highly organized, patterned, and predictable. Additionally, a carefully developed procedure is a powerful tool for solving routine tasks.

Strategic competence is the ability to formulate, present, and solve mathematical problems with sub-indicators used to determine whether students (a) understand the problem, (b) present it in various forms (numerical, symbolic, verbal, or graphic), (c) select the appropriate formula, approach, or method, and (d) check the correctness of the answer obtained. The fundamental characteristic required during the problem-solving process is flexibility. This is developed by expanding the knowledge needed to solve non-routine problems.

Adaptive reasoning refers to thinking logically about the relationship between concepts and situations and the ability to reflect, explain, and justify specific issues. The sub-indicators for this variable are whether students can (a) construct conjectures, (b) provide reasons or evidence to prove a statement is true, (c) conclude, (d) check the validity of an argument, and (e) find patterns in a mathematical phenomenon.

Productive disposition is related to the tendency to develop constructive habits, see mathematics as reasonable, useful, meaningful, and valuable, as well as to have confidence and perseverance during learning. The indicators for this variable are if the students are (a) enthusiastic, (b) do not give up easily, (c) confident, (d) highly curious, and (e) willing to share.

Students with a highly productive disposition tend to be able to develop certain capabilities in terms of conceptual understanding, procedural fluency, strategic competence, and adaptive reasoning. Meanwhile, those with self-proficiency regarding conceptual understanding, procedural fluency, strategic competence, and adaptive reasoning tend to develop a productive disposition. The development of these five components of self-capabilities needs to be carried out in an integrated manner [11].

Based on the earlier description, self-proficiency is interpreted as the ability or competence and skill possessed by a person in terms of understanding as well as employing procedures and strategies in performing certain activities with enthusiasm, persevering, being confident, highly curious, and possessing the willingness to share.

### 2.4. Cooperative Mathematics Learning Model: Team-Assisted Individualization Type

The cooperative learning model was implemented by prioritizing the use of student groups. The principle that should be adhered to in relation to this categorization is that students in a particular group must possess a heterogeneous ability level. If necessary, they should be from different races, cultures, and ethnicities. Consideration should also be given to gender equality [50].

Cooperative learning teaches the relevance of cooperation and collaboration to improve the students' social skills. Ref. [51] stated that "its models not only help these learners to understand difficult concepts rather, it also aids them to develop cooperative and critical thinking skills as well as the ability to assist friends". The cooperative learning model improves the students' mathematical problem-solving, communication, and self-proficiency abilities.

One type of cooperative learning is team-assisted individualization (TAI). It is described as a group learning method in which a more capable student acts as an assistant who is in charge of individually helping those who are less advanced in a group. In this case, the teacher only plays the role of a facilitator and mediator in the teaching and learning process. It is enough for them to create a conducive learning environment for the students [52]. The TAI learning model motivates learners to help one another in their groups and to create enthusiasm in the competitive system by prioritizing individual roles without sacrificing cooperative aspects.

This learning model has eight stages in its implementation: (1) placement test, (2) teams, (3) teaching groups, (4) creative student, (5) team study, (6) fact test, (7) team score and team recognition, as well as (8) whole-class unit. This cooperative learning model also has stages that bring about aspects of mathematical problem-solving, communication, and self-proficiency abilities with problem-solving strategies for all students in the class. Each component in the TAI type of the cooperative learning model benefits teachers, learners, and upper and lower groups who work together to complete academic tasks. The students who grasped a particular concept more quickly are responsible for helping those who are still integrating the new concept into their system of knowledge. This enables all students to develop their abilities and skills.

### 3. Methods

This quantitative study employed a quasi-experimental model [53]. Research was carried out on students enrolled in the undergraduate program of statistics in the odd semester of the 2019/2020 academic year who took the operations research course at the Faculty of Mathematics and Natural Sciences, at Hasanuddin University in Makassar City, Indonesia. Purposive sampling was adopted based on the criteria of selecting new students taking the course for the first time, and 92 of them were used as the study sample.

Of the total 96 students from two parallel classes of operations research courses, four students were eliminated for taking this course for the second time. Thus, 92 of them were used as the study sample, which consisted of 50 students in the first class; and 42 students in the second class, from which had been eliminated four students. Two large classes were randomly set for experimental and control groups as samples, where one consisting of 50 students was used as an experimental group with the team-assisted individualization (TAI) type of cooperative learning model. Meanwhile, the other class consisting of 42 students was used as the control group with a conventional learning model. In applying the team-assisted individualization type of the cooperative learning model, the division of small study groups is set at 5–6 students. The demographic characteristics of students as research subjects are heterogeneous based on gender, ethnicity, religion, the origin of residence (coming from Makassar city or outside Makassar city), economic status, and social status.

Data collection techniques involve using test instruments to analyze the improvement of mathematical problem-solving, communication, and self-proficiency abilities with the TAI type of cooperative learning model. Relevant information was obtained from the mathematical prior knowledge (MPK) test, mathematical problem solving (MPS), mathematical communication (MC), and self-proficiency (SPr) ability increase tests, observation sheets, interview guidelines, learning tools (student worksheets), and the teaching program unit (TPU). Measurement of validity was undertaken before the test was carried out using data collection instruments to obtain valid data. The researcher consulted with the lecturer of the operations research course in this study to fulfill the theoretical validity of the instrument. Moreover, we conducted a reliability test regarding the agreement of scores between several raters, including among assessors who were just learning to rate. Therefore, it involves assessors with experience teaching operations research, as well as assessors who have less experience teaching operations research and less familiarity with this course [54,55].

These were further analyzed using parametric and non-parametric statistical evaluations with prerequisite tests for data normality and homogeneity of variance. Various statistical tests were carried out on the dimensions of class groups or mathematical prior knowledge levels using statistical analysis instruments, such as the independent sample and the Kolmogorov–Smirnov, Levene, Mann–Whitney, One-Way ANOVA, Two-Way ANOVA, Kruskal–Wallis, and Tamhane tests. Data processing was performed with statistical analysis instruments using the SPSS version 22 application.

## 4. Results and Discussion

### 4.1. Mathematical Prior Knowledge of Students

The grouping of students based on the mathematical prior knowledge (MPK) test results comprises those with high, medium, and low abilities [56,57]. This test was carried out to obtain an overview of their initial mathematical abilities by calculating the mean and standard deviation. Table 2 shows the students' initial mathematical ability scores where the mean, standard deviation, as well as minimum and maximum values of the experimental and control classes are not significantly different. Therefore, it was concluded that the two groups have essentially the same MPK. Before the equivalence evaluation was conducted, the data normality and homogeneity of variance tests were performed using an independent sample analysis for the experimental and control classes.

**Table 2.** Descriptive Statistic of the Mathematical Prior Knowledge of Students.

| Statistic | Experimental | Control | Total |
|:---:|:---:|:---:|:---:|
| $n$ | 50 | 42 | 92 |
| $\bar{x}$ | 18.15 | 16.67 | 17.47 |
| $s$ | 5.75 | 4.94 | 5.34 |
| Max. | 28 | 27 | 28 |
| Min. | 10 | 7 | 7 |

Table 3 shows that the normality test results for the experimental and control classes are asym-sig < 5%; therefore, it was concluded that the data are not normally distributed. The non-normality of the MPK data distribution indicates that it is not necessary to test the homogeneity of variance. A non-parametric statistical analysis in accordance with the Mann–Whitney test was conducted to evaluate the difference in the mean of the two independent sample groups. Table 3 shows that the Mann–Whitney U test results are sig. > 5%. It was concluded that the MPK had no significant difference in the experimental and control classes. The determination of learning groups in this study can be selected randomly, which strengthens the justification of the grouping.

**Table 3.** Kolmogorov–Smirnov and Mann–Whitney U tests for Mathematical Prior Knowledge of Students.

| Observation Groups | $n$ | $\overline{x}$ | Kolmogorov–Smirnov Test | | Mann–Whitney U Test | |
|---|---|---|---|---|---|---|
| | | | KS | Asym. Sig | Z | Sig. (2-Tailed) |
| Experimental | 50 | 18.15 | 0.12 | 0.00 | −0.94 | 0.34 |
| Control | 42 | 16.67 | 0.22 | 0.00 | | |

Furthermore, the classification of students into three groups is based on their mathematical prior knowledge level, including those with high, medium, and low abilities. This grouping is based on the mean and standard deviation of the combined experimental and control classes with the formulation $\overline{x} \pm s$. Therefore, $\overline{x} + s = 22.81$ and $\overline{x} - s = 12.13$ are obtained. The sample distribution based on the mathematical prior knowledge of students in each learning group is shown in Table 4.

**Table 4.** Distribution of Samples based on the Classification of Mathematical Prior Knowledge Levels and Learning Groups.

| Mathematical Prior Knowledge (MPK) Levels | Learning Groups | | Total |
|---|---|---|---|
| | Experimental | Control | |
| High | 9 | 8 | 17 |
| Medium | 33 | 27 | 60 |
| Low | 8 | 7 | 15 |
| Total | 50 | 42 | 92 |

*4.2. Mathematical Problem-Solving Ability Analysis*

Mathematical problem solving (MPS) ability analysis was based on descriptive statistics in terms of mean ($\overline{x}$) and standard deviation ($s$) for pre- and post-test scores, as well as normalized gain (N-gain) [7]. Based on Table 5, the experimental class experienced a higher increase in mathematical problem-solving than the control. This is proven by the N-gain in each experimental class at low, medium, high, and combined MPK levels. Furthermore, the mean post-test of the experimental class shows a higher score than the control at low, medium, high, and combined ability levels.

Based on the learning model applied, the higher the student's MPK level, the more significant the increase in mathematical problem-solving ability. It descriptively shows that in each MPK classification, the increase in the mathematical problem-solving ability of the experimental class is greater than the control. This is indicated by the difference in N-gain in both classes for each MPK level.

Normality and homogeneity of variance tests were carried out before the mean difference analysis of data on the increase in mathematical problem-solving ability. This was based on the learning model with mathematical prior knowledge level. The Kolmogorov–Smirnov test was carried out to evaluate the normality of the data on the increase in

mathematical problem-solving ability in both classes. Additionally, Levene's test is used to evaluate the homogeneity of variance.

**Table 5.** Descriptive Statistics from Data on the Increase in Students' Mathematical Problem-Solving Ability.

| MPK Level | Des. Stat. | Experimental | | | | Control | | | |
|---|---|---|---|---|---|---|---|---|---|
| | | Pre-Test | Post-Test | N-Gain | *n* | Pre-Test | Post-Test | N-Gain | *n* |
| High | $\bar{x}$ | 18.33 | 37.22 | 0.71 | 9 | 14.25 | 26.33 | 0.39 | 8 |
| | *s* | 3.50 | 4.66 | | | 0.89 | 1.55 | | |
| Medium | $\bar{x}$ | 13.88 | 25.22 | 0.36 | 33 | 12.37 | 18.85 | 0.20 | 27 |
| | *s* | 2.70 | 3.13 | | | 4.17 | 2.45 | | |
| Low | $\bar{x}$ | 4.25 | 21.13 | 0.41 | 8 | 1.25 | 9.17 | 0.18 | 7 |
| | *s* | 2.27 | 1.91 | | | 0.51 | 1.35 | | |
| Total | $\bar{x}$ | 13.14 | 26.73 | 0.43 | 50 | 10.87 | 18.66 | 0.23 | 42 |
| | *s* | 2.79 | 3.30 | | | 3.50 | 2.21 | | |

Note: the maximum (ideal) score is 45.

Table 6 shows that the data normality test results of the increase in mathematical problem-solving ability for all mathematical prior combined, high, medium, and low knowledge levels in the experimental and control classes are asym. sig. > 5%; hence, the acquired data are normally distributed. The homogeneity of variance test with Levene's test was carried out to determine whether or not the variance of the scores measured in the two classes have similarities. The Levene's test results at the combined, high, and medium MPK levels show sig. > 5%; it was concluded that the data on the increase in students' mathematical problem-solving ability has a homogeneous variance. The results of the low MPK level are sig. < 5%; it implies that the data has an inhomogeneous variance. A mean difference test was conducted on the increase in mathematical problem-solving abilities in the experimental and control classes for the combined, high, and medium MPK levels with the T-test using the independent sample test. The low MPK level with T'-test was carried out using the Mann–Whitney test.

**Table 6.** Kolmogorov–Smirnov and Levene's Tests of Data on the Increase in Mathematical Problem-Solving Ability.

| MPK Level | Learning Group | *n* | Kolmogorov–Smirnov Test | | Levene's Test | | | |
|---|---|---|---|---|---|---|---|---|
| | | | KS | Asym. Sig. | F | df$_1$ | df$_2$ | Sig. |
| Combined | Experimental | 50 | 0.10 | 0.23 | 4.08 | 1 | 90 | 0.07 |
| | Control | 42 | 0.09 | 0.28 | | | | |
| High | Experimental | 9 | 0.25 | 0.27 | 0.39 | 1 | 15 | 0.55 |
| | Control | 8 | 0.13 | 0.20 | | | | |
| Medium | Experimental | 33 | 0.06 | 0.17 | 2.03 | 1 | 58 | 0.16 |
| | Control | 27 | 0.10 | 0.20 | | | | |
| Low | Experimental | 8 | 0.13 | 0.20 | 5.21 | 1 | 13 | 0.04 |
| | Control | 7 | 0.22 | 0.20 | | | | |

Table 7 shows the T- and T'-test results with sig. < 5%; therefore, it was concluded that there is a mean difference in the increased mathematical problem-solving ability in the experimental and control classes for the combined, high, medium, and low MPK levels. Based on these tests, which are supported by descriptive statistics, it was concluded that there is an increase in the mathematical problem-solving ability of the experimental class, which is perceived to be better than the control at the combined, high, medium, and low MPK levels. A one-way ANOVA test was conducted to test the mean difference relating to

the experimental class's increase in mathematical problem-solving ability. The normality prerequisite test is shown in Table 6, with the homogeneity of variance test carried out at the mathematical prior knowledge level using Levene's test.

**Table 7.** T- and T'-tests on the Difference Test of the Increase in Mathematical Problem-Solving Ability in Experimental and Control Classes.

| MPK Level | df | T | Z | Sig. (One-Tailed) |
|-----------|-----|------|------|-------------------|
| Combined | 90 | 7.29 | - | 0.00 |
| High | 15 | 4.12 | - | 0.00 |
| Medium | 58 | 5.79 | - | 0.00 |
| Low | 13 | - | 3.66 | 0.00 |

Table 8 shows Levene's test results with sig. > 5%; therefore, it was concluded that the data on the increase in mathematical problem-solving ability in the experimental class has a homogeneous variance among the three MPK levels. Table 9 shows the mean difference test results for the increase in mathematical problem-solving ability in the experimental class using the one-way ANOVA test with sig. > 5%. It was concluded that there was no mean difference in the increase in students' mathematical problem-solving ability in the experimental class based on the mathematical prior knowledge levels. Due to the fact that there is no significant mean difference in the increase in mathematical problem solving ability at the three MPK levels, it is necessary to discontinue the post hoc test. An interaction test was rather carried out between the learning model and the MPK level toward the increase in mathematical problem solving ability.

**Table 8.** Levene's Test of Data on the Increase in Mathematical Problem-Solving Ability for the Experimental Class at the Three MPK Levels.

| F | $df_1$ | $df_2$ | Sig. |
|------|--------|--------|------|
| 1.94 | 2 | 47 | 0.82 |

**Table 9.** One-Way ANOVA Test on Data on the Increase in Mathematical Problem-Solving Ability based on MPK Level in the Experimental Class.

| Source of Variance | Sum of Square | d.f. | Mean Square | F | Sig. |
|--------------------|---------------|------|-------------|------|------|
| Between Group | 0.09 | 2 | 0.05 | 1.24 | 0.30 |
| Within Group | 1.74 | 47 | 0.04 | | |
| Total | 1.83 | 49 | | | |

The interaction between the learning model and the prior knowledge level and its impact on the increase in mathematical problem-solving ability was examined using a two-way ANOVA test. This evaluation starts with the prerequisite test to determine normality and homogeneity. The normality test is shown in Table 6, and its results are normally distributed. Furthermore, the homogeneity of variance test with Levene's test is shown in Table 10 and sig. > 5%. It was concluded that the data on the increase in students' mathematical problem-solving ability based on the learning model, and the MPK level has a homogeneous variance.

**Table 10.** Levene's Test on Data on the Increase in Mathematical Problem-Solving Ability based on Mathematical Prior Knowledge Level and Learning Model.

| F | $df_1$ | $df_2$ | Sig. |
|------|--------|--------|------|
| 0.01 | 2 | 89 | 0.99 |

The two-way ANOVA test was carried out to fulfill the assumptions that need to be met, namely the sampling technique should use independent random sampling, and the data has to be normally distributed (with or without the same mean, but must have similar variance). Based on Table 11, the F-test results on the learning model show sig. < 5%; therefore, it was concluded that learning has a significant effect on the increase in mathematical problem-solving ability. The F-test results at the MPK level show sig. < 5%, and it was concluded that there is a significant difference in the increase in mathematical problem-solving ability related to the learning model and the MPK level. The interaction test between the learning model and the MPK level shows sig. > 5%, and it was concluded that there is no interaction between the learning model and MPK level. It has an insignificant impact on the increase in mathematical problem-solving ability.

**Table 11.** Two-Ways ANOVA Test on the Increase in Mathematical Problem-Solving Ability based on Learning Model and Mathematical Prior Knowledge Level.

| Source | Type III Sum of Squares | df | Mean Square | F | Sig. |
|---|---|---|---|---|---|
| Corrected Model | 1.64 | 5 | 0.33 | 11.98 | 0.00 |
| Intercept | 4.30 | 1 | 4.30 | 156.62 | 0.00 |
| Learning Model | 0.90 | 1 | 0.90 | 32.79 | 0.00 |
| MPK | 0.09 | 2 | 0.04 | 1.60 | 0.03 |
| Learning Model and MPK | 0.02 | 2 | 0.01 | 0.36 | 0.70 |
| Error | 2.36 | 86 | 0.03 | | |
| Total | 13.57 | 92 | | | |

$R^2 = 0.41$ (Adj-$R^2 = 0.41$).

Due to the fact that there is an increase in mathematical problem-solving ability that differs significantly based on the MPK level, the post hoc test was conducted. The essence is to determine the mean difference in the increased mathematical problem-solving ability in the experimental class according to the MPK level using the Tamhane test. The assumption of homogeneity of variance is shown in Table 8; therefore, the Tamhane test uses data on the increase in mathematical problem-solving ability based on the MPK level. In accordance with Table 12 regarding the pairing test, the MPK-level group has sig. < 5%. This shows that the mean difference in the increased mathematical problem-solving ability occurred in pairs of the MPK-level groups. It was also concluded that based on the Tamhane test, the increase in mathematical problem-solving in the high MPK level is better than both the medium and low levels. The medium MPK level is also better than the low one. Figure 3 shows that there is no interaction between the experimental and control classes' learning model and the mathematical prior knowledge levels solving problems.

**Table 12.** Tamhane Test on the Increase in Mathematical Problem-Solving Ability between Pairs of Mathematical Prior Knowledge Level Groups.

| MPK (I) | MPK (J) | Mean Difference | Sig. |
|---|---|---|---|
| High | Medium | 0.29 | 0.00 |
| | Low | 0.55 | 0.00 |
| Medium | High | −0.29 | 0.00 |
| | Low | 0.26 | 0.00 |
| Low | High | −0.55 | 0.00 |
| | Medium | −0.26 | 0.00 |

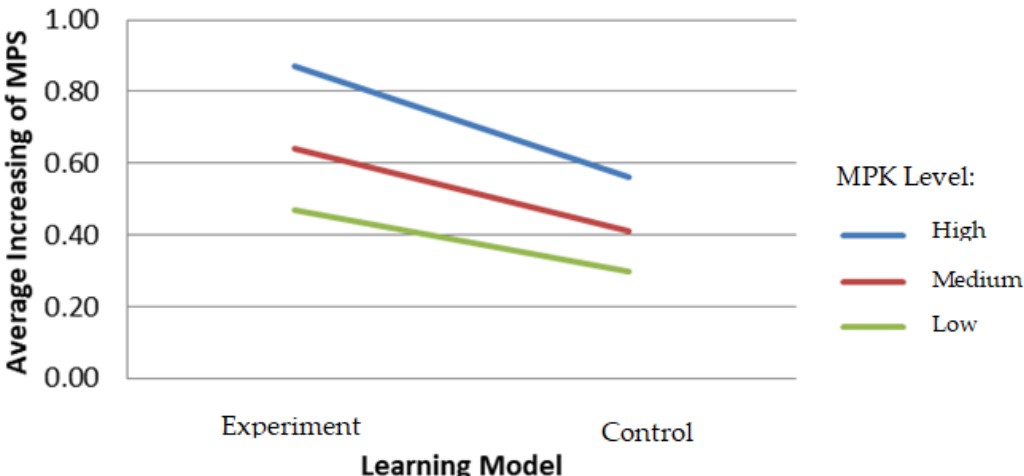

**Figure 3.** Interaction between Learning Model and MPK level on the Increase in Mathematical Problem-Solving Ability.

Based on the previous explanation, it was concluded that the increase in the problem-solving ability of students taught with the team-assisted individualization type of cooperative learning model is better than those lectured with the conventional method for all mathematical prior knowledge levels. Asides from being influenced by the learning model, prior mathematical knowledge contributes to differences in the students' increased mathematical problem-solving ability. Therefore, the TAI type of cooperative learning model involves the formation of small heterogeneous groups with different mathematical prior knowledge levels to help other students in need. Observations on mathematical problem-solving show that students have enhanced abilities in understanding and exploring problems, determining strategies, and evaluating mathematical problem-solving. The application of this learning model in the experimental class has a positive effect on improving problem-solving abilities, especially for students with low MPK level, as well as those with high to medium levels. This is because there is more acceptance for students with a lower MPK level in the TAI class than in than the control class. Students with medium and high MPK levels have a relatively higher increase in problem-solving abilities because they are also motivated to carry out team-assisted activities with others and with students with lower MPK levels. This was performed to enable lecturers to organize or arrange appropriate learning steps, thereby increasing the students' mathematical solving ability [7,10,12,13,33,40].

*4.3. Mathematical Communication Ability Analysis*

Mathematical communication (MC) ability analysis was performed using pre- and post-tests and N-gain. A descriptive statistical review for the mean ($\bar{x}$) and standard deviation ($s$) of the pre- and post-tests and N-gain values is based on the learning model and mathematical prior knowledge level. Based on Table 13, the increase in mathematical communication ability in the experimental class is higher than in control. This is evident in the higher N-gain value realized from the experimental class compared to the control. Moreover, this is supported by the mean post-test versus the pre-test.

The mean difference test was carried out on the increase in mathematical communication ability based on the learning model and MPK level. It commenced with the prerequisite evaluations, such as normality and homogeneity of variance tests. Based on Table 14, the normality test results of the data on the increase in mathematical communication ability at all MPK levels in the experimental and control classes show asym. sig. > 5%; therefore, it is normally distributed. The homogeneity of variance test results show sig. > 5% at the combined high and medium MPK levels, and the data are presumed to have a homogeneous variance. The low MPK level has sig. < 5%; the data are presumed to have a heterogeneous variance. The mean difference analysis was carried out with the T-test using

the independent sample analysis and the Mann–Whitney test to determine the increase in mathematical communication ability.

**Table 13.** Descriptive Statistics from Data on the Increase in Mathematical Communication Ability.

| MPK Level | Des. Stat. | Experimental | | | | | Control | | | |
|---|---|---|---|---|---|---|---|---|---|---|
| | | Pre-Test | Post-Test | N-Gain | *n* | | Pre-Test | Post-Test | N-Gain | *n* |
| High | $\overline{x}$ | 18.33 | 35.57 | 0.64 | 9 | | 16.33 | 25.39 | 0.32 | 8 |
| | *s* | 2.30 | 1.12 | | | | 1.07 | 1.31 | | |
| Medium | $\overline{x}$ | 13.88 | 27.55 | 0.44 | 33 | | 11.51 | 19.37 | 0.23 | 27 |
| | *s* | 2.60 | 4.43 | | | | 4.99 | 4.27 | | |
| Low | $\overline{x}$ | 4.26 | 19.63 | 0.38 | 8 | | 3.01 | 8.71 | 0.14 | 7 |
| | *s* | 2.25 | 2.37 | | | | 0.47 | 1.55 | | |
| Total | $\overline{x}$ | 13.14 | 27.73 | 0.46 | 50 | | 11.01 | 18.74 | 0.23 | 42 |
| | *s* | 2.50 | 3.80 | | | | 4.10 | 3.58 | | |

Note: the maximum (ideal) score is 45.

**Table 14.** Kolmogorov–Smirnov and Levene's Tests on Data for the Increase in Mathematical Communication Ability.

| MPK Level | Learning Group | *n* | Kolmogorov–Smirnov Test | | Levene's Test | | | |
|---|---|---|---|---|---|---|---|---|
| | | | KS | Asym. Sig. | F | $df_1$ | $df_2$ | Sig. |
| Combined | Experimental | 50 | 0.12 | 0.07 | 2.12 | 1 | 90 | 0.15 |
| | Control | 42 | 0.11 | 0.27 | | | | |
| High | Experimental | 9 | 0.30 | 0.27 | 0.03 | 1 | 15 | 0.97 |
| | Control | 8 | 0.17 | 0.20 | | | | |
| Medium | Experimental | 33 | 0.11 | 0.20 | 3.07 | 1 | 58 | 0.15 |
| | Control | 27 | 0.10 | 0.20 | | | | |
| Low | Experimental | 8 | 0.19 | 0.20 | 4.10 | 1 | 13 | 0.01 |
| | Control | 7 | 0.28 | 0.14 | | | | |

Table 15 shows the T- and T'-test results with sig. < 5%; therefore, it was concluded that there is a mean difference in the increased mathematical communication ability in the experimental and control classes for combined, high, medium, and low MPK levels. Based on the fact that descriptive statistics supported these tests, it was concluded that the increased mathematical communication ability in the experimental class is better than the control at all MPK levels. Furthermore, a one-way ANOVA test was conducted to determine the mean difference in the increased mathematical communication ability in the experimental class. The normality prerequisite test is shown in Table 14, while the homogeneity of variance test is carried out at the mathematical prior knowledge level in accordance with Levene's test.

**Table 15.** T- and T'-tests on the Difference Test of the Increase in Mathematical Communication Ability in the Experimental and Control Classes.

| MPK Level | Df | T | Z | Sig. (One-Tailed) |
|---|---|---|---|---|
| Combined | 90 | 4.85 | - | 0.00 |
| High | 15 | 4.32 | - | 0.00 |
| Medium | 58 | 13.04 | - | 0.00 |
| Low | 13 | - | 2.66 | 0.02 |

Table 16 shows Levene's test results with sig. < 5%; therefore, it was concluded that the increased mathematical communication ability in the experimental class has a heterogeneous variance.

**Table 16.** Levene's Test of Data on the Increase in Mathematical Communication Ability for the Experimental Class at the Three MPK Levels.

| F | $df_1$ | $df_2$ | Sig. |
|---|---|---|---|
| 4.32 | 2 | 47 | 0.02 |

Table 17 shows the mean difference test results of the increase in mathematical communication ability in the experimental class using the one-way ANOVA test with sig. < 5%; therefore, it was concluded that there is a mean difference based on the mathematical prior knowledge level. The next step is to perform a post hoc test to determine the mean difference in the experimental class according to the MPK level using the Tamhane test.

**Table 17.** One-Way ANOVA Test of Data on the Increase in Mathematical Communication Ability based on MPK Levels in the Experimental Class.

| Source of Variance | Sum of Square | d.f. | Mean Square | F | Sig. |
|---|---|---|---|---|---|
| Between Group | 0.74 | 2 | 0.37 | 31.73 | 0.00 |
| Within Group | 0.55 | 47 | 0.01 | | |
| Total | 1.29 | 49 | | | |

Table 18 shows that the pair test for the MPK-level groups has a sig. < 5%. This implies that the mean difference in the increased mathematical communication ability in the experimental class occurs in all pairs of the MPK-level groups. Based on the Tamhane test, there are differences at the high, medium, and low MPK levels. The high MPK level is better than the medium and low MPK level. In addition, the medium MPK level is better than the low one.

**Table 18.** Tamhane's Test on the Increase in Mathematical Communication Ability Between Pairs of Mathematical Prior Knowledge Level Groups in the Experimental Class.

| MPK (I) | MPK (J) | Mean Difference | Sig. |
|---|---|---|---|
| High | Medium | 0.25 | 0.00 |
| | Low | 0.60 | 0.00 |
| Medium | High | −0.25 | 0.00 |
| | Low | 0.35 | 0.00 |
| Low | High | −0.60 | 0.00 |
| | Medium | −0.35 | 0.00 |

Furthermore, the interaction between the learning model and the MPK level on the increase in students' mathematical communication ability was observed using the two-way ANOVA test, which is preceded by data normality and homogeneity of variance prerequisite assessments. The normality test is shown in Table 14, and the acquired data are normally distributed. The homogeneity of variance, in accordance with Levene's test, is shown in Table 19 to produce sig. > 5%. It was concluded that the data has a homogeneous variance.

**Table 19.** Levene's Test on Data for the Increase in Mathematical Communication Ability based on Mathematical Prior Knowledge Level and Learning Model.

| F | $df_1$ | $df_2$ | Sig. |
|---|---|---|---|
| 0.03 | 2 | 89 | 0.97 |

Based on Table 20 regarding the two-way ANOVA test results, the learning model has a sig. < 5%; therefore, it significantly affects the increase in mathematical communication ability. The MPK level also has a sig. < 5%, meaning it significantly affects the increase in mathematical communication ability. In accordance with these results, it was concluded that there is a significant difference in the increased mathematical communication ability in the learning model and the MPK level. The interaction test was used to obtain sig. > 5%, meaning there is no connection between the increase in mathematical communication ability with respect to the learning model and MPK level.

**Table 20.** Two-Ways ANOVA Test on the Increase in Mathematical Communication Ability based on Learning Model and Mathematical Prior Knowledge Level.

| Source | Type III Sum of Squares | df | Mean Square | F | Sig. |
|---|---|---|---|---|---|
| Corrected Model | 0.88 | 5 | 0.18 | 7.07 | 0.00 |
| Intercept | 3.73 | 1 | 3.73 | 149.26 | 0.00 |
| Learning Model | 0.00 | 2 | 0.00 | 0.05 | 0.00 |
| MPK | 0.25 | 1 | 0.25 | 10.04 | 0.00 |
| Learning Model and MPK | 0.07 | 2 | 0.04 | 1.45 | 0.24 |
| Error | 2.15 | 86 | 0.03 | | |
| Total | 12.75 | 92 | | | |

$R^2 = 0.29$ (Adj-$R^2 = 0.25$).

Figure 4 shows no interaction between the experimental and control classes and the MPK level on the increase in students' mathematical communication ability.

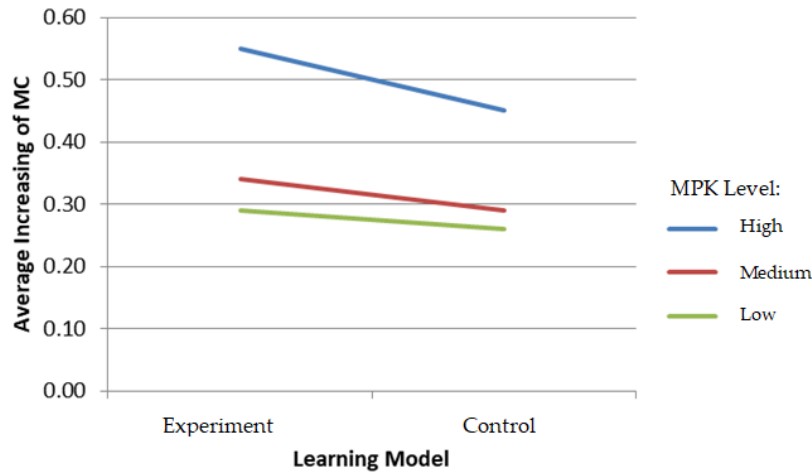

**Figure 4.** Interaction between Learning Model and MPK Level on the Increase in Mathematical Communication Ability.

Overall, it was concluded that the increase in mathematical communication ability of students taught with the TAI type of the cooperative learning model is better than those lectured with the conventional method for each mathematical prior knowledge level. Furthermore, the results show that the higher the MPK level, the greater the increase in

mathematical communication ability. At each MPK level, the increase in mathematical communication ability in the experimental class is greater than in the control. In general, the increased mathematical communication observations showed an increased ability to express numerical ideas orally or in writing and to demonstrate and present them visually. Furthermore, it is good for understanding, interpreting, as well as evaluating mathematical ideas orally. Students are able to use terms, arithmetic notation, and their structures to present ideas and describe the relationships and situations of the model being studied. This learning model can improve students' mathematical communication skills because there is a more dominant interaction than in conventional learning models. The formation of small groups in this model forces students with low MPK levels to communicate mathematically to solve the given mathematical problems. They can adapt the mathematical communication process to students with higher communication skills, so they are motivated, and their abilities improve. In addition, students with communication skills in medium and high MPK levels undergo an analysis process of students whose MPK levels are below to improve their mathematical communication ability further [20–22,28,42].

### 4.4. Self-Proficiency Analysis

Self-proficiency (SPr) analysis is preceded by descriptive statistics, namely the mean ($\bar{x}$) dan standard deviation ($s$) of the initial, final, and N-gain values based on the learning model and MPK level. Table 21 shows that the mean increase (N-gain) of this variable in the experimental class is higher than the control regarding MPK levels. This depicts that the increase in self-proficiency of students taught with the TAI type of cooperative learning model is at a medium level (medium N-gain category) on all MPK. Moreover, those lectured with the conventional learning method are at a lower level on all MPK levels.

**Table 21.** Descriptive Statistics from Student Self-Proficiency Data.

| MPK Level | Des. Stat. | Experimental | | | | Control | | | |
|---|---|---|---|---|---|---|---|---|---|
| | | Initial | Final | N-Gain | *n* | Initial | Final | N-Gain | *n* |
| High | $\bar{x}$ | 118.2 | 145.4 | 0.41 | 9 | 117.7 | 121.9 | 0.06 | 8 |
| | $s$ | 11.2 | 9.7 | | | 8.2 | 7.1 | | |
| Medium | $\bar{x}$ | 99.5 | 132.4 | 0.38 | 33 | 97.9 | 118.5 | 0.24 | 27 |
| | $s$ | 10.8 | 8.1 | | | 6.9 | 7.7 | | |
| Low | $\bar{x}$ | 87.1 | 121.7 | 0.35 | 8 | 86.6 | 106.8 | 0.21 | 7 |
| | $s$ | 7.9 | 6.2 | | | 5.1 | 8.1 | | |
| Total | $\bar{x}$ | 100.9 | 133.0 | 0.38 | 50 | 99.7 | 117.2 | 0.21 | 42 |
| | $s$ | 10.5 | 8.1 | | | 6.9 | 7.7 | | |

Note: the maximum (ideal) score is 185.

Furthermore, the mean difference test is carried out on the increase in self-proficiency based on the learning model and MPK level. Prerequisite assessments, including normality and homogeneity of variance tests, were initially conducted. Table 22 shows that the normality test on the increase in self-proficiency for the combined, high, and low mathematical prior knowledge levels in both experimental and control classes obtained an asym. sig. > 5%, meaning the data are normally distributed. At the medium MPK level for the experimental class, asym sig. < 5% was obtained, depicting that the data were abnormally distributed, although the reverse was the case for the control. The homogeneity of variance test results shows a sig. > 5% at the combined, high, and low MPK levels, thereby causing the data to have a homogeneous variance. For the medium MPK level, the sig. < 5%, meaning the data has a heterogeneous variance. The mean difference evaluation on the increase in self-proficiency for the experimental and control classes at the combined, high, medium, and low MPK levels were carried out using the independent sample test, T- and T'-tests, and the Mann–Whitney test.

**Table 22.** Kolmogorov–Smirnov and Levene's Tests on Data for the Increase in Self-Proficiency.

| MPK Level | Learning Group | n | Kolmogorov–Smirnov Test | | Levene's Test | | | |
|---|---|---|---|---|---|---|---|---|
| | | | KS | Asym. Sig. | F | $df_1$ | $df_2$ | Sig. |
| Combined | Experiment | 50 | 0.08 | 0.20 | 0.00 | 1 | 90 | 0.07 |
| | Control | 42 | 0.08 | 0.11 | | | | |
| High | Experiment | 9 | 0.16 | 0.20 | 3.09 | 1 | 15 | 0.11 |
| | Control | 8 | 0.17 | 0.20 | | | | |
| Medium | Experiment | 33 | 0.15 | 0.04 | 7.09 | 1 | 58 | 0.01 |
| | Control | 27 | 0.09 | 0.20 | | | | |
| Low | Experiment | 8 | 0.22 | 0.20 | 2.20 | 1 | 13 | 0.11 |
| | Control | 7 | 0.30 | 0.20 | | | | |

Table 23 shows that the mean difference test result obtained is sig. < 5% for the increase in self-proficiency in the combined group and each MPK level. Descriptive statistics support the mean value and the two-mean difference test results. Table 21 shows that the increase in the self-proficiency of the experimental class is higher than the control.

**Table 23.** T- and T'-tests on the Difference Test for the Increase in Self-Proficiency in the Experimental and Control Classes.

| MPK Level | df | T | Z | Sig. (One-Tailed) |
|---|---|---|---|---|
| Combined | 90 | 7.82 | - | 0.00 |
| High | 15 | 2.56 | - | 0.02 |
| Medium | 58 | - | −6.90 | 0.00 |
| Low | 13 | 2.95 | - | 0.01 |

Furthermore, a one-way ANOVA test was carried out to examine the difference in the increased self-proficiency in the experimental class at each MPK level. The prerequisite assessment for the afore-mentioned analysis is the normality test which does not fulfill the requirements in Table 22, meaning the homogeneity of variance test was not performed. The mean difference test in the increased self-proficiency was determined using the Kruskal–Wallis test. Table 24 shows the result obtained is asymp. sig. < 5%, and it was concluded that there is a mean difference in the increase in self-proficiency in the three MPK groups of the experimental class. This is because it is proven to be different in each MPK group, then a post hoc test was conducted. The essence was to evaluate the mean difference in the increased mathematical communication ability in the experimental class according to the MPK level using the Tamhane test.

**Table 24.** Kruskal–Wallis Test of Data on the Increase in Self-Proficiency for Experimental Class at the Three MPK Levels.

| $\chi^2$ | df | Asym. Sig. (One-Tailed) |
|---|---|---|
| 24.36 | 2 | 0.03 |

Based on Table 25, the pair test for the MPK-level groups obtained a sig. < 5%. This shows that the mean difference of the increase in self-proficiency in the experimental class occurred in all pairs of MPK levels. Based on the Tamhane test, there are differences in the increased self-proficiency at the high, medium, and low MPK levels. The high MPK level is

better than the medium and low MPK levels. In addition, the medium MPK level is better than the low one.

**Table 25.** Tamhane Test on the Increase in Self-Proficiency between Pairs of Mathematical Prior Knowledge Level Groups in the Experimental Class.

| MPK (I) | MPK (J) | Mean Difference | Sig. |
|---|---|---|---|
| High | Medium | 5.23 | 0.00 |
| | Low | 9.17 | 0.00 |
| Medium | High | −5.23 | 0.00 |
| | Low | 3.94 | 0.00 |
| Low | High | −9.17 | 0.00 |
| | Medium | −3.94 | 0.00 |

The next analysis is to observe the interaction between the experimental and control classes at all MPK levels. This was unable to be carried out using a two-way ANOVA test with a parametric statistical approach. The reason is that the prerequisite tests were not fulfilled, where the normality of the data on the increase in self-proficiency in the control class for the medium MPK level is abnormally distributed. Therefore, the interaction test was carried out descriptively, as shown in Figure 5.

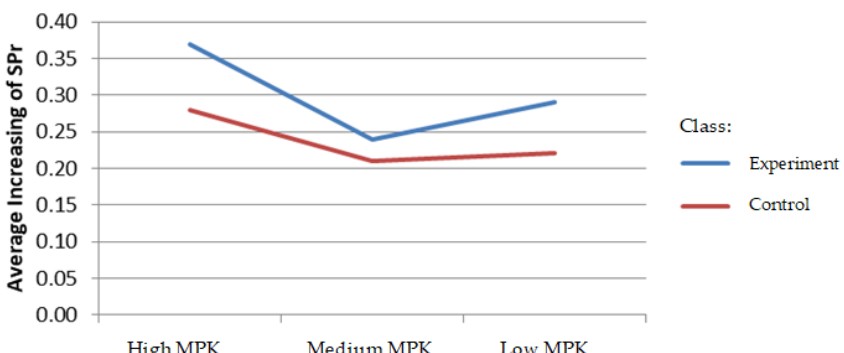

**Figure 5.** Interaction between Learning Model and MPK Level on the Increase in Self-proficiency.

Based on Figure 5, the increase in students' self-proficiency at all MPK levels in the experimental class is higher than in control. The graph proves that there is no interaction in the experimental and control classes as well as at the MPK level related to the increase in Self-proficiency. This is evident in the non-parallel lines depicting the two learning models shown on the graph. Meanwhile, the difference in the mean at each MPK level is similar.

Overall, it was concluded that the increase in self-proficiency of students taught with the TAI type of the cooperative learning model is better than those lectured with the conventional approach for each mathematical prior knowledge level. Furthermore, the learning model and MPK level factors show that the higher the MPK level, the greater the increase in self-proficiency. At each MPK level, the increase in self-proficiency in the experimental group is greater than in the control class. The application of the TAI type of the cooperative learning model, which is proven to improve mathematical problem-solving and communication abilities, is related to increasing self-proficiency. More dominant interaction and acceptance in small groups further increased self-proficiency. In the observation process, it was found that students were more proficient in restating learning concepts, could classify whether or not criteria were met for forming specific models, and provided applicable examples of concepts outside the learning topic. However, students with low MPK levels are somewhat less proficient in presenting ideas in other forms of mathematical representation and developing the necessary or sufficient conditions for conception, although students with medium and high MPK levels have these improvements. Moreover, the observations

showed an increase in procedural ability to apply, utilize, select, and predict the results of procedural fluency in the knowledge of procedural accuracy, as well as its application flexibly, accurately, and efficiently. Meanwhile, at low MPK levels, they tend not to be proficient in modifying, perfecting, and developing procedures, even though students with moderate and high MPK levels increase these abilities.

Analysis of the increase in aspects of strategic competence found an increase in students' ability to understand mathematical problems and present problem-solving in numerical, symbolic, verbal, and graphic forms. In addition, the improved ability is seen to have formulas, approaches, and selections of suitable methods in problem-solving. However, students with low MPK levels show that they have less-developed abilities to re-examine the correctness of mathematical problem-solving answers, although students with moderate and high MPK levels are proficient in these abilities. Related to the adaptive reasoning ability, it was found that the students were able to formulate conjectures, provide reasons or evidence for proving problem-solving, and conclude solved cases. Students with low and moderate MPK levels have low abilities to examine the validity of an argument for solving mathematical problems and finding patterns in a mathematical phenomenon, while students with high MPK levels have well-developed abilities in these areas. For productive disposition ability, students at low and medium MPK levels have good abilities related to enthusiasm, perseverance, self-confidence, high curiosity, and willingness to share. In contrast, students with high MPK levels tend to have low abilities in this aspect. They score well in enthusiasm for learning, self-confidence, and willingness to share, but do not score as well in the areas of perseverance, great curiosity, and willingness to share to students in their groups. For students who have a high MPK level, it is suspected that they have several weaknesses due to their tendency to be more proficient compared to their fellow students who have a lower MPK level. Related to improving these aspects, it shows the characteristics of each level of MPK and the different strengths and weaknesses associated with each bracket. Furthermore, the increase in self-proficiency is inseparable from the increase in mathematical problem solving and communication; therefore, it is necessary to examine the relationship between mathematical problem solving, communication, and self-proficiency [11,12,47–49,58,59].

### 4.5. Correlation between Mathematical Problem Solving, Communication, and Self-Proficiency

Normality tests using Kolmogorov–Smirnov test were carried out as a prerequisite to identify the correlation between mathematical problem solving, communication, and self-proficiency with a parametric statistical approach.

Table 26 shows that the three variables are normally distributed. Furthermore, the correlation between mathematical problem solving, communication, and self-proficiency was tested using the Pearson correlation because it fulfills the assumption of a normal distribution.

**Table 26.** Kolmogorov–Smirnov Test on Mathematical Problem Solving, Communication, and Self-proficiency.

| Variable | KS | Asym. Sig. |
|---|---|---|
| Problem Solving | 0.13 | 0.06 |
| Communication | 0.12 | 0.08 |
| Self-Proficiency | 0.13 | 0.09 |

Table 27 shows that a positive and significant correlation exists between problem solving with communication, problem solving, and self-proficiency, and communication with self-proficiency. The correlation between problem solving and communication is $r_p = 0.83$ with sig. < 5%. An absolutely strong and significant positive correlation exists between the two variables. This shows that students with the high mathematical problem-solving ability also have great mathematical communication skills and vice versa. The theoretical study by [60] is in line with this finding.

**Table 27.** Pearson Correlation between Mathematical Problem Solving, Communication, and Self-Proficiency.

| Among Variables | $r_p$ | Sig. |
|---|---|---|
| Problem ↔ Communication | 0.83 | 0.00 |
| Problem ↔ Self-proficiency | 0.81 | 0.00 |
| Communication ↔ Self-proficiency | 0.67 | 0.01 |

The correlation between problem solving and self-proficiency is $r_p = 0.81$ with sig. $< 5\%$. An extremely strong and significant positive correlation exists between the two variables. This shows that students with the high problem-solving ability also have increased self-proficiency, and vice versa. The result is supported by [11,59], which shows that the problem-based learning model can develop mathematical proficiency. Therefore, students with exceptional mathematical problem-solving ability also have good self-proficiency [58]. The correlation between communication and self-proficiency is $r_p = 0.67$ with sig. $< 5\%$. A positive correlation at the moderate and significant level between these two variables shows that students with high mathematical communication ability also have increased self-proficiency skills, and vice versa.

The correlation between mathematical problem solving, communication, and self-proficiency is evident in the sub-indicators. Individuals usually tend to find a way around a mathematical problem; therefore, it is important to ascertain how the various representations relate to one another, their similarities, and their differences. The students' conceptual understanding level relates to the richness and breadth of the connections they can make. There is a correlation between mathematical problem solving, communication, and self-proficiency. This is proven by the sub-indicators, namely, presenting a mathematical problem in various forms, such as numeric, symbolic, verbal, or graphic. The correlation obtained is $r_p = 0.375$, although its coefficient is in the low category. The statistical test results confirm that the correlation between mathematical problem solving, communication, and self-proficiency is significant. This shows that students with high mathematical problem solving and communication also have great self-proficiency and vice versa. The acquired result is in line with [61], which demonstrated that mathematical problem solving and communication are two complementary processes.

### 4.6. Implementation of the Team-Assisted Individualization (TAI) Type of the Cooperativelearning Model

The TAI type of the cooperative learning model has eight stages in its implementation: (1) placement test, (2) teams, (3) teaching groups, (4) creative student, (5) team study, (6) fact test, (7) team score and recognition, and (8) whole-class units. First of all, the lecturer gives the students a placement test to determine their ability to initiate the learning process. Heterogeneous groups consisting of five students each are formed, and the course material is briefly delivered before the students are given assignments.

Furthermore, the lecturer emphasizes the fact that the group determines the success of each student. External motivation is needed in the form of encouragement from group members. Students study together by working on assignments from the worksheets given to their groups. In the next stage, the lecturer, or those with exceptional academic abilities as peer tutors, offers individual assistance to learners who need help. At this phase, social cohesion is strengthened, because each member actively plays a role and wants to be part of the group. Therefore, group members are dependent on one another. An interactive session can support learning according to the students' cognitive development [28,62].

The teacher and students implement the learning model in the experimental class based on direct observations. Learning was carried out over the course of 10 meetings, each executed for 3-semester credit systems. Therefore, each of the meetings lasted for 150 min. In the first meeting, the topic was solving linear programming problems using

the graphical method. The students analyzed alternative solutions graphically using the following material:

Pay attention to the following
Standard form → Minimize $Z = 5x_1 + 3x_2$
With obstacles

(1)  $2x_1 + x_2 - x_3 + x_4 = 3$
(2)  $x_1 + x_2 - x_5 + x_6 = 2$

**Phase I**

(0)  $Z = x_4 + x_6 \rightarrow -Z = -x_4 - x_6$

Basic variables: $x_4$ & $x_6$
Non-basic variables: $x_1, \ x_2, x_3, \ and \ x_5$

(0)  $-Z + x_4 + x_6 = 0$
(1)  $2x_1 + x_2 - x_3 + x_4 = 3$

________________________-

$-Z - 2x_1 - x_2 + x_3 + x_6 = -3$
(2)  $x_1 + x_2 - x_5 + x_6 = 2$

________________________-

$-Z - 3x_1 - 2x_2 + x_3 + x_5 = -5$

Problem expansion

(0)  $-Z - 3x_1 - 2x_2 + x_3 + x_5 = -5$
(1)  $2x_1 + x_2 - x_3 + x_4 = 3$
(2)  $x_1 + x_2 - x_5 + x_6 = 2$

| Basic Variable | Equation | Z | $x_1$ | $x_2$ | $x_3$ | $x_4$ | $x_5$ | $x_6$ | Right Section | Minimal Ratio |
|---|---|---|---|---|---|---|---|---|---|---|
| | (0) | −1 | −3 | −2 | 1 | 0 | 1 | 0 | −5 | |
| $x_4$ | (1) | 0 | 2 | 1 | −1 | 1 | 0 | 0 | 3 | 3/2 |
| $x_6$ | (2) | 0 | 1 | 1 | 0 | 0 | −1 | 1 | 2 | 2 |

| Basic Variable | Equation | Z | $x_1$ | $x_2$ | $x_3$ | $x_4$ | $x_5$ | $x_6$ | Right Section | Minimal Ratio |
|---|---|---|---|---|---|---|---|---|---|---|
| | (0) | −1 | 0 | −1/2 | −1/2 | 3/2 | 1 | 0 | −1/2 | |
| $x_1$ | (1) | 0 | 1 | 1/2 | −1/2 | 1/2 | 0 | 0 | 3/2 | 3 |
| $x_6$ | (2) | 0 | 0 | 1/2 | 1/2 | −1/2 | −1 | 1 | 1/2 | 1 |

| Basic Variable | Equation | Z | $x_1$ | $x_2$ | $x_3$ | $x_4$ | $x_5$ | $x_6$ | Right Section | Minimal Ratio |
|---|---|---|---|---|---|---|---|---|---|---|
| | (0) | −1 | 0 | 0 | 0 | 1 | 0 | 1 | 0 | |
| $x_1$ | (1) | 0 | 1 | 0 | −1 | 1 | 1 | −1 | 1 | |
| $x_2$ | (2) | 0 | 0 | 1 | 1 | −1 | −2 | 2 | 1 | |

| Basic Variable | Equation | Z | $x_1$ | $x_2$ | $x_3$ | $x_4$ | $x_5$ | $x_6$ | Right Section | Minimal Ratio |
|---|---|---|---|---|---|---|---|---|---|---|
| | (0) | −1 | 0 | 0 | 0 | 1 | 0 | 1 | 0 | |
| $x_1$ | (1) | 0 | 1 | 0 | −1 | 1 | 1 | −1 | 1 | |
| $x_2$ | (2) | 0 | 0 | 1 | 1 | −1 | −2 | 2 | 1 | |

**Phase II**

(0)  $-Z = -5x_1 - 3x_2$
(1)  $x_1 - x_3 + x_5 = 1$

(2)  $x_2 + x_3 - 2x_5 = 1$

(0)  $-Z + 5x_1 + 3x_2 = 0$

(2)  $3x_2 + 3x_3 - 6x_5 = 3 \ldots \ldots x3$

$$\overline{\hspace{6cm}}\,\text{-}$$

$-Z + 5x_1 - 3x_3 + 6x_5 = -3$

(1)  $5x_1 - 5x_3 + 5x_5 = 5 \ldots \ldots x5$

$$\overline{\hspace{6cm}}\,\text{-}$$

$-Z + 2x_3 + x_5 = -8$

Phase II Tabulation

| Basic Variable | Equation | Z | $x_1$ | $x_2$ | $x_3$ | $x_5$ | Right Section |
|---|---|---|---|---|---|---|---|
| | (0) | −1 | 0 | 0 | 2 | 1 | −8 |
| $x_1$ | (1) | 0 | 1 | 0 | −1 | 1 | 1 |
| $x_2$ | (2) | 0 | 0 | 1 | 1 | −2 | 1 |

Conclusion: It is optimal with a Minimum and Maximum of −8 and 8, respectively with $(x_1, x_2) = (1, 1)$.

The implementation of learning activities is in accordance with what is planned in the teaching program unit, in this case, at meeting one (LPU 1). Initially, the students reluctantly followed the learning model in the experimental class because it was relatively different from the conventional method. The lecturer played an active role in directing and guiding them to be able to adapt to this approach. The learning process was carried out in small groups, and gradually, the students started to develop a positive attitude by respecting one another, exhibiting democratic attitudes, respecting differences, taking responsibility, as well as establishing togetherness and cooperative behavior. Therefore, with this strategy, students were able to learn how to solve problems together.

The students were given simple questions with different answer options, which left them confused as to the right response to select. However, students became highly curious in understanding that every response is relevant and needs to be delivered according to thoughts. When students presented group results, it was usually associated with a response provided with high enthusiasm. The class atmosphere becomes more fun, and students actively participate in group discussions by properly completing the questionnaire provided by the lecturer. Through the tutor's guidance in the previously assigned problems, the students were able to solve linear programming questions using the two-phase method as shown in Figure 6.

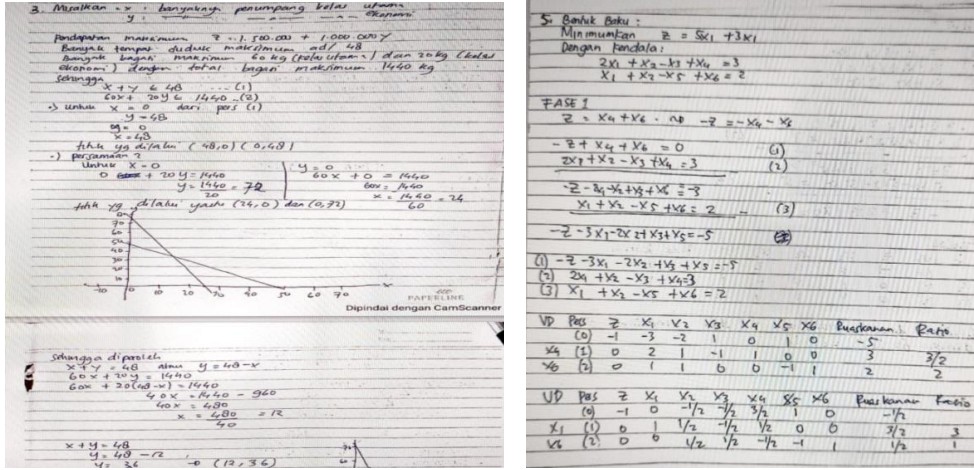

**Figure 6.** Answer Sheet for Working on Linear Program Problems with the Two-Phase Method by One of the Students.

In determining the two-phase method used to solve linear programming problems by creating artificial variables, students are directed to re-identify other ways to solve these questions. The determination of this topic is in the form of an equation, although it does not yet have a basis, therefore the artificial variable needs to be added. With the use of the two-phase method, the value of $c_j = 0$, $\forall_j$, and the cost coefficient for the artificial variable is $-1$ (maximum problem). The lecturer then directs students to determine the two-phase method's basic concept by asking several questions: "How can one use this approach to solve linear programming problems? Is the afore-mentioned question already in the form of an equation, but does not contain the basis, thereby leading to the addition of an artificial variable?"

It seems that the students provided diverse answers, although after they were made to understand that their responses were relevant, it boosted their confidence, and they became excited.

Subsequently, the lecturer scored the group work and awarded both those that performed brilliantly and those who performed less successfully with the following "titles": "OK group," "EXCELLENT group,", etc. Finally, the lecturer briefly presented the material at the end of the meeting with problem-solving strategies for all students in the class.

## 5. Conclusions, Implications, Recommendations, and Limitations

Based on data analysis and comprehensive findings, it was concluded that the increase in mathematical problem-solving ability in the experimental class was better than the control. There was no interaction between the students among experiment and control classes. This also includes those at each mathematical prior knowledge (MPK) level in relation to increasing mathematical problem-solving ability. Furthermore, the increase in mathematical communication ability in the experimental class is higher than in control. There was no interaction between the students among experimental and control classes, as well as those at each MPK level related to increased mathematical communication ability. The increase in self-proficiency in the experimental class is better than in the control. There is no interaction between the students in the experimental and control classes, including those at each MPK level associated with increasing self-proficiency. There is a significant correlation between students' mathematical problem-solving, communication, and self-proficiency abilities.

The advantages of the team-assisted individualization (TAI) type of the cooperative learning model are that it applies a combination of individual and group learning, the material presented is easily understood by the students, their enthusiasm is high, and the learning atmosphere is conducive. Meanwhile, the disadvantage is that the adopted steps are lengthy, thereby requiring a long time to solve. This bored students with high abilities because they needed to adjust to those who took longer to understand the concepts.

This study implies that the TAI type of the cooperative learning model is appropriate for students who take the operations research course as an alternative to develop mathematical problem-solving, communication, and self-proficiency abilities. A sufficient correlation exists between mathematical communication and problem-solving abilities, implying that its development is relevant. Furthermore, the absence of significant interaction between the learning model and students' MPK on mathematical problem-solving, communication, and self-proficiency abilities indicates that learning does not interact with students' initial mathematical abilities but with those who receive TAI learning. This learning model can be applied to lectures not only in the field of statistics or mathematics but also in other departments whose curriculum includes the operations research course, such as accounting and finance, marketing, construction, operations and supply chain management, etc.

This study offers several recommendations as follows: the team-assisted individualization (TAI) type of the cooperative learning model is an alternative for lecturers in implementing the operations research course, specifically for improving mathematical problem-solving skills and communication, including self-proficiency for students. The TAI learning model, adopted in small groups consisting of heterogeneous members, encourages

students to be actively involved in learning activities. The obstacles encountered during its implementation were overcome by assigning students to study the material first according to the teaching program unit independently planned because a comprehensive understanding makes it easier and faster to solve the problem within a short time. Moreover, this learning method is expected to be a trigger to shape the construction of students' thinking in the future in problem-solving, communication, and self-proficiency abilities to face other aspects of life.

The present study has several limitations. Firstly, it only carried out its analysis using a quantitative approach. Previous research [1,12] employed a mixed-method procedure only limited to the aspect of achievement on the variables. Meanwhile, this study comprehensively examines more complex variables specifically for the context of increase but failed to use a qualitative approach. Future studies can examine this matter qualitatively, such as studying students' concrete solutions from a didactic perspective, such as problem-solving strategies in the learning process, the obstacles faced, and related matters for improving aspects of mathematical problem-solving abilities, communication, and self-proficiency. Secondly, if there are more than 50 observations per group in the experimental and control groups, this statistical analysis instrument is inadequate. Subsequent research that has a subject of observation of more than 50 students can use correspondence and hierarchical cluster (CHIC) [63].

Thirdly, it only examines the increase of these three variables in the TAI type of the cooperative learning model, which has several weaknesses. Further research can examine these three variables in various relevant learning models. It can also be tested in other subjects, different education levels in the university, and under certain conditions to determine the reliability of the increase in these variables, specifically self-proficiency as a new attribute. Fourthly, it is necessary to explain that the subject of this research is students at state universities in Indonesia, where the selection process to be accepted into universities is rigorous and full of competition between prospective students from cities or regions around the university area or another area. Therefore, students' ability at this university tends to be higher than the normal population in general. Therefore, further research can use a sample more representative of the normal population, not only at state universities, but also from a representative population from various other universities.

**Author Contributions:** Conceptualization, G.M.T., B.N., A.B.H., and P.G.H.; methodology, G.M.T., B.N., A.B.H., and P.G.H.; software, G.M.T. and P.G.H.; validation, G.M.T., B.N.; formal analysis, G.M.T., A.B.H., and P.G.H.; investigation, G.M.T., B.N., A.B.H., and P.G.H.; resources, G.M.T., B.N., A.B.H., and P.G.H.; data curation, G.M.T. and P.G.H.; writing—original draft preparation, G.M.T., B.N., A.B.H., and P.G.H.; writing—review and editing, G.M.T., B.N., A.B.H., and P.G.H.; visualization, G.M.T., A.B.H., and P.G.H.; supervision, G.M.T. and B.N.; project administration, G.M.T. and P.G.H.; funding acquisition, G.M.T., B.N., A.B.H., and P.G.H. All authors have read and agreed to the published version of the manuscript.

**Funding:** This study received no external funding.

**Institutional Review Board Statement:** The research has ethical clearance by the Department of Statistics, Hasanuddin University, with the Letter of Statement for Ethical Approval number 9119/UN4.11.7/PT.0105/2022.

**Informed Consent Statement:** Informed consent was obtained from all subjects, namely students involved in this study.

**Data Availability Statement:** The data presented in this study are available on request from the corresponding author. The data are not publicly available due to privacy reasons.

**Acknowledgments:** The authors are grateful to the students of the operations research course as the observation subjects; and to the Department of Statistics, Hasanuddin University Faculty of Mathematics and Natural Sciences, for the Ethical Clearance review. The authors are also grateful to the reviewers and editorial board of Education Sciences for their constructive and thoughtful comments.

**Conflicts of Interest:** The authors declare no conflict of interest.

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
