# Peer review of "Team-Assisted Individualization Type of the Cooperative Learning Model for Improving Mathematical Problem Solving, Communication, and Self-Proficiency: Evidence from Operations Research Teaching"

_education, doi:10.3390/educsci12110825_

Round 1

Reviewer 1 Report

The paper is devoted to the research in the abilities of university students in the field of mathematical problem-solving, development of mathematical communication skills and self-proficiency abilities and competencies. It was realized a proof of Team-Assisted Individualization type of the Cooperative and conventional Learning Models in the groups - experimental and control. The results of the research were discussed. I have some recommendations and comments:

 1. Page 1, row 10 The sample of 50 and 42 students were from undergraduate study program of statistics? It is possible to put here this additional information.

2. Page 3, row 119 It is possible to add some references from the new literature. Some journals offered special issues and papers about Mathematical Problem Solving (see https://www.mdpi.com/journal/sustainability/special_issues/math_education_problem-solving or https://www.mdpi.com/2071-1050/12/23/10113).

3. Page 5, row 207 Figure 2 The red text in green boxes is bad readable. I recommend changing graphics and colour.

4.  Page 20, row 718 The text has bad graphical form, please correct.

5. Page 23, row 820 I don’t understand what it means “There is no interaction between the students in both classes “. Why is this sentence repeated in the rows 823, 826?   

6. Page 24, row 860 I think that the limitation was also number of students in experimental and control group. If the number will be more greater than 50 students, the it is possible to use various statistical methods for example C.H.I.C (see for example some English papers in https://sites.univ-lyon2.fr/asi/10/pub/ASI10_ISBN_978-2-9562045-3-4_NUMERIQUE2020s.pdf or https://sites.univ-lyon2.fr/asi/11/pub/ASI11_ISBN_978-2-9562045-5-8_NUMERIQUE2021.pdf).

7. Page 24, row 864 It is possible to add some comments to the further research of the solutions of students described on page 22, row 775 – Figure 6. It is very valuable to analyse concrete solutions of students from didactical point of view – for example students´ solving strategies, possible obstacles in understanding of mathematics notions and so on.

Author Response

Dear Reviewer 1

We are grateful for the constructive and thoughtful comments on our research paper so that the research paper is more qualified to publish in Education Sciences and contributes more to the literature. Therefore, we have revised our research paper with responses to the comments attached to this reply. Once more, we thank you very much.

Our best regards,
Authors.

Reviewer 2 Report

Dear authors and editor,

I have completed the revision of the submission of your paper titled: "Team-Assisted Individualization Type of Cooperative Learning Model for Improving Mathematical Problem Solving, Communication, and Self-Proficiency: Evidence from Operations Research Teaching”.

I found the manuscript to be engaging and well-structured and the methods and results clearly and concisely presented. I consider the conclusions adequate, in relation to the analysis and methods that have been employed. In addition, the conclusions are well linked with the theoretical framework that has been presented. Overall, this piece of research is very interesting and useful for educational research.

Below, I note a few aspects that should be improved to strengthen the manuscript:

1. Although the introduction and review of the bibliography are adequate, I strongly recommend that references to some works be included due to their relevance and influence in the teaching of mathematics, and especially in problem solving. Among the references that the authors should add are:

Bell, A. (1996). Problem-solving approaches to algebra: Two aspects. En N. Bernardz, C. Kieran y L. Lee (Eds.), Approaches to algebra. Perspectives to research and teaching (pp. 167-187). Dordretch, The Netherlands: Kluwer Academic Publishers.

Nesher, P., Hershkovitz, S., & Novotna, J. (2003). Situation model, Text base and what else? Factors affecting problem solving. Educational Studies in Mathematics, 52(2), 151–176. https://doi.org/10.1023/A:1024028430965

Schoenfeld, A. H. (1992). Learning to think mathematically: Problem solving, metacognition, and sense making in mathematics. In D. A. Grouws (Ed.), Handbook of research on mathematics teaching and learning (pp. 334–370). National Council of Teachers of Mathematics.

In particular, some concepts such as those developed in lines 128-136 (skills necessary to deal with successful problem solving) are already reflected in some of the references mentioned. Thus, for example, Bell (1996) focuses on problem solving beyond what he calls the narrow sense of the term, which would refer to problem solving by formulating and solving equations. Thus, the author speaks of a broad sense of the term to accommodate more general mathematical processes and activities of generalization and functional modelling, for example.

2. The methods presented and used in the investigation seem totally adequate to me. However, I would like the authors to clarify the robustness of the ANOVA, especially in relation to the differences measured in each section and the size of the sample taken in consideration. Perhaps it has been clarified in the text, but during my reading I have not found it clear enough.

Author Response

Dear Reviewer 2

We are grateful for the constructive and thoughtful comments on our research paper so that the research paper is more qualified to publish in Education Sciences and contributes more to the literature. Therefore, we have revised our research paper with responses to the comments attached to this reply. Once more, we thank you very much.

Our best regards,
Authors.

Reviewer 3 Report

The problem situation and necessity of the study should be explained in more detail.

The originality of this study and its contribution to the field should be discussed.

The purpose and sub-objectives of the research should be clearly stated.

There are serious deficiencies in the method part of the research. How were the experimental and control groups of the research formed? Detailed information should be given. In addition, information about the demographic characteristics of the participants should be given. The model of the research should be detailed. There is also a need for detailed information about data collection tools. Validity and reliability analyzes should be mentioned in detail. The applications made in the experimental and control groups should be explained in detail. Has attention been paid to the ethical procedures of the research? What was done for this purpose?

The results obtained should be correlated with the literature. The discussion part of the study was particularly weak in terms of being supported by relevant research.

Author Response

Dear Reviewer 3

We are grateful for the constructive and thoughtful comments on our research paper so that the research paper is more qualified to publish in Education Sciences and contributes more to the literature. Therefore, we have revised our research paper with responses to the comments attached to this reply. Once more, we thank you very much.

Our best regards,
Authors.

Round 2

Reviewer 3 Report

corrections are appropriate